# Managing the Dual Nature of Iron to Preserve Health

**DOI:** 10.3390/ijms24043995

**Published:** 2023-02-16

**Authors:** Laura Silvestri, Mariateresa Pettinato, Valeria Furiosi, Letizia Bavuso Volpe, Antonella Nai, Alessia Pagani

**Affiliations:** 1Regulation of Iron Metabolism Unit, Division of Genetics and Cell Biology, San Raffaele Scientific Institute, 20132 Milan, Italy; 2School of Medicine, Vita-Salute San Raffaele University, 20132 Milan, Italy

**Keywords:** IRP1, IRP2, iron deficiency, iron overload, transferrin, TFR1, hepcidin, ferroportin, BMP-SMAD, erythroferrone, TFR2, HFE, HJV, TMPRSS6

## Abstract

Because of its peculiar redox properties, iron is an essential element in living organisms, being involved in crucial biochemical processes such as oxygen transport, energy production, DNA metabolism, and many others. However, its propensity to accept or donate electrons makes it potentially highly toxic when present in excess and inadequately buffered, as it can generate reactive oxygen species. For this reason, several mechanisms evolved to prevent both iron overload and iron deficiency. At the cellular level, iron regulatory proteins, sensors of intracellular iron levels, and post-transcriptional modifications regulate the expression and translation of genes encoding proteins that modulate the uptake, storage, utilization, and export of iron. At the systemic level, the liver controls body iron levels by producing hepcidin, a peptide hormone that reduces the amount of iron entering the bloodstream by blocking the function of ferroportin, the sole iron exporter in mammals. The regulation of hepcidin occurs through the integration of multiple signals, primarily iron, inflammation and infection, and erythropoiesis. These signals modulate hepcidin levels by accessory proteins such as the hemochromatosis proteins hemojuvelin, HFE, and transferrin receptor 2, the serine protease TMPRSS6, the proinflammatory cytokine IL6, and the erythroid regulator Erythroferrone. The deregulation of the hepcidin/ferroportin axis is the central pathogenic mechanism of diseases characterized by iron overload, such as hemochromatosis and iron-loading anemias, or by iron deficiency, such as IRIDA and anemia of inflammation. Understanding the basic mechanisms involved in the regulation of hepcidin will help in identifying new therapeutic targets to treat these disorders.

## 1. Introduction

Iron is one of the most abundant elements on Earth, but it is poorly bioavailable because ferrous iron (Fe^2+^), the soluble form primarily used in biological processes, is rapidly oxidized to ferric iron (Fe^3+^) in aerobic environments. Due to its unique oxidoreductive properties and its propensity to exchange electrons, iron is essential for life. Although Fe^3+^ is insoluble at neutral pH, it can be efficiently dissolved at acidic pH. Thus, to take up or acquire iron, organisms have evolved two mechanisms: the acidification of their environment and the reduction of Fe^3+^ to Fe^2+^.

In living cells, iron can be found in prosthetic groups, such as iron–sulfur clusters (ISCs) and heme, a complex of iron and protoporphyrin, or coordinated with oxygen in iron-containing enzymes. In this way, iron participates in a variety of biological functions such as energy production, oxygen transport, deoxyribonucleotide production, and deoxyribonucleic acid (DNA) replication and repair. Iron’s propensity to accept or donate electrons makes this metal crucial to several biological processes. In humans, most of the body’s iron is contained in hemoglobin (Hb) in erythrocytes: its deficiency causes anemia due to decreased Hb synthesis and red blood cell production. However, severe hypoferremia may also impact other cells and tissues due to the reduced synthesis of iron-containing enzymes and proteins, crucial for cell functions. On the other hand, the chemical reactivity of iron makes it potentially highly toxic: ferrous iron (Fe^2+^), which is unstable under aerobic conditions, triggers the Fenton reaction, generating reactive oxygen species that can damage proteins, membranes, and DNA. To avoid the potential damaging effects of iron, this metal is always bound to proteins for its uptake and transport and its storage in cells. In humans, iron overload causes cell and tissue damage due to the appearance of unbound iron, which can result in organ failure and even death in the most severe cases. For these reasons, organisms have evolved mechanisms to tightly regulate iron homeostasis and prevent both its deficiency and excess.

## 2. Iron Absorption

Dietary iron, in the form of inorganic iron and heme, is absorbed by enterocytes in the duodenum (Figure 1A). Inorganic, non-heme Fe^3+^ is solubilized due to the low pH of the duodenal lumen and reduced to Fe^2+^ by the ferrireductase Duodenal Cytochrome b (DCytB) of the apical membrane of enterocytes. It is then absorbed through divalent metal transporter 1 (DMT1), a H+/metal symporter that uses the high duodenal H+ concentration to drive transmembrane iron import [1]. The mechanism of heme iron absorption by enterocytes is less well known and is likely mediated by Heme Carrier Protein 1 (HCP1), a folate/proton symporter that is thought to play a role in intestinal heme and folate absorption. HCP1 is highly expressed in the duodenum and has been shown to function as a low-affinity heme transporter [2,3,4]. Heme entry into the cell is also facilitated by a vesicular transport system activated by the binding of heme to the apical side of enterocytes [5]. After its import, heme is degraded by heme oxygenase (HO), which releases Fe^2+^ into the cell.

Once in the cytosol, Fe^2+^ can be utilized or exported to the bloodstream by the sole iron exporter ferroportin (FPN1) (Figure 1A), a ubiquitous protein abundant in enterocytes, macrophages, and hepatocytes. FPN1 is characterized by a transmembrane N-lobe and C-lobe that form a hydrophilic channel through which Fe2+ is exported from the cell. The export of Fe^2+^ is coupled to its extracellular oxidation to Fe^3+^, mediated by the plasma membrane multicopper ferroxidase hephaestin (HEPH) expressed in the duodenum (Figure 1A). Other ferroxidases with a similar function are ceruloplasmin (CP), expressed in macrophages and hepatocytes, and zyklopen (ZP), mainly expressed in the placenta [7]. The machinery that regulates dietary iron absorption is coordinated by hypoxia-inducible factor 2α (HIF2), a transcription factor whose levels are under the control of oxygen tension and iron [8,9]. Decreased iron concentrations in enterocytes lead to the transcriptional upregulation of *DCytB*, *DMT1*, and *FPN1*, whose expression in this cell type is regulated by HIF2α [10]. This represents a mechanism that counteracts body iron deficiency by increasing dietary iron absorption. Interfering with this mechanism by using HIF2α inhibitors was shown to be effective in reducing iron burden in diseases characterized by iron overload due to increased dietary iron absorption, such as hereditary hemochromatosis or ineffective erythropoiesis [9].

Excess intracellular iron that is not exported or utilized forms the so-called “labile iron pool” and is safely stored in ferritin cages, which consist of 24 subunits of ferritin H (FTH) and L (FTL) (Figure 1A). While FTL is inactive, FTH has ferroxidase activity and converts Fe^2+^ to insoluble Fe^3+^ to store it in ferritin cages in a chemically inactive form [11]. Iron is delivered to ferritins via direct metal-mediated protein–protein interactions through the iron chaperones Poly (rC)-binding proteins 1 and 2 (PCBP1 and 2), which bind cytosolic Fe^2+^ [12]. Conversely, iron stored in ferritin is released back into the cytosol by the cargo nuclear receptor coactivator 4 (NCOA4) via a degradation process known as ferritinophagy [13], a process that is activated under conditions of cellular iron deficiency [14] (Figure 1A).

Interestingly, low iron availability also affects DNA metabolism via NCOA4. The nuclear localization of this cargo receptor is stabilized in iron deficiency to inhibit DNA replication, protecting the cell from replication stress, genome instability, and death [15].

## 3. Iron Transport

In the blood, insoluble Fe^3+^ is bound with high affinity by transferrin (TF), a glycoprotein produced by hepatocytes that allows iron solubilization and safe transport to virtually all cells and tissues (Figure 1B). TF is characterized by a bilobed structure, with each lobe, the N- and C-lobes, binding one Fe^3+^ atom. Under physiological conditions, approximately one-third of circulating TF is saturated with iron and circulates mainly as monoferric TF, whereas under iron-loading conditions, TF is mainly diferric. Interestingly, the occupancy of the N- or C-lobe by Fe^3+^ in monoferric TF influences the regulation of iron homeostasis and erythropoiesis [16]. Diferric-TF, or holo-TF, is the form of iron used by virtually all cells in the body, although the main consumers are erythroid cells, which use this metal to synthesize hemoglobin in a process known as erythropoiesis. Diferric-TF binds transferrin receptor 1 (TFR1), a homodimeric type II transmembrane glycoprotein localized on the cell surface, and is internalized via an endocytic process (Figure 1B). TFR1 forms a homodimer that binds a TF molecule to each of its subunits with different affinities depending on TF saturation. TFR1 binds diferric-TF with a higher affinity than monoferric or iron-free (apo) forms. The binding of diferric-TF to TFR1 activates the internalization of the ligand–receptor complex by clathrin-mediated endocytosis. The acidic pH of the endosome facilitates the release of Fe^3+^ from TF, whereas the apo-TF-TFR1 complex is recycled back to the cell surface. Here, the neutral pH of the extracellular environment promotes TF dissociation from TFR1 [17]. Fe^3+^ in the endosome is reduced to Fe^2+^ by the ferrireductase six-transmembrane epithelial antigen of prostate 3 (STEAP3) and then exported to the cytosol by the H^+^/metal symporter DMT1 (Figure 1B). Fe^2+^ is rapidly consumed, mainly by mitochondria, or stored in ferritin cages in the cytosol.

Interestingly, diferric-TF can also bind transferrin receptor 2 (TFR2), which is highly homologous to TFR1 but has a different function. TFR1 is ubiquitously expressed, with reduced expression when the intracellular iron concentration is elevated, and is involved in cellular iron uptake, while TFR2 is mainly expressed in hepatocytes, erythroid cells, and osteoblasts, is stabilized on the cell surface by diferric-TF, and has mainly regulatory functions [18] (Table 1).

When the serum iron concentration, usually maintained in the range of 10–30 μM, exceeds the buffering capacity of TF, non-transferrin-bound iron (NTBI) appears and accumulates in an uncontrolled manner in hepatocytes and parenchymal cells in various organs, leading to toxicity. NTBI uptake is mediated mainly by transporters that, under physiological conditions, transport other metals, such as ZIP14 and ZIP8, which transport zinc, and L-type and T-type calcium channels (LTCCs/TTCCs) [19].

## 4. Iron Utilization

The main cellular sites of iron utilization are the mitochondria, the organelles primarily involved in the synthesis of heme and iron–sulfur clusters (ISCs) (Figure 1C). Although the outer mitochondrial membrane (OMM) is permeable to ions and small metabolites, transport across the inner membrane (IMM) requires carrier proteins. Several mechanisms are potentially involved in iron transport to the OMM, such as fluid-phase endocytosis, contact with lysosomes or endosomes, metallochaperone transport, and labile iron pool (LIP) transport. In addition, the “kiss and run” interaction, a process mainly described in erythroblasts, characterizes the contact between iron-loaded endosomes and mitochondria [20,21]. It consists of a transient and direct interorganelle interaction in which a large amount of iron is delivered to mitochondria to support heme synthesis for hemoglobin and red blood cell production. The direct transfer of Fe^2+^ prevents potential ROS production by the highly reactive metal in the cytosol. Two proteins regulate iron transport through the IMM: mitoferrin 1 (MFRN1), which is highly expressed in erythroblasts, and mitoferrin 2 (MFRN2), which plays an important role in other cell types (Figure 1C). MFRN1 is part of a protein complex essential for heme synthesis, and its functional inactivation leads to severe anemia in mice due to impaired hemoglobin and erythrocyte production [22]. In mitochondria, iron can be used for heme and ISC synthesis or, if not utilized, stored in mitochondrial ferritin (MTFT), a protein expressed in various tissues and cell types, such as the kidney, brain, and sperm. It is also expressed under certain pathological conditions, such as in erythroblasts of patients with sideroblastic anemia. MTFT, provided with ferroxidase activity, can store iron similarly to its cytosolic counterparts [23], thus protecting the organelle from oxidative stress.

### 4.1. Heme Biosynthesis

Heme, a complex of Fe^2+^ and protoporphyrin IX, plays a crucial role in oxygen transport and energy production. It is the prosthetic group of hemoglobin and myoglobin, oxidoreductases and oxidases of respiratory complexes I, II, III, and IV, and other proteins. Heme also regulates signal transduction and gene expression. Heme synthesis is a complex process that requires the coordinated activity of enzymes localized in both the mitochondria and cytosol (Figure 1C). The first step, which occurs in the mitochondria, involves the condensation of glycine (which is imported into the organelle by SLC25A38) and succinyl-CoA (an intermediate of the TCA cycle) by the enzyme 5-aminolevulinate synthase (ALAS) to form delta-aminolevulinic acid (ALA) (Figure 1C). Of the two ALAS genes, ALAS1 is ubiquitously expressed, while iron-regulated ALAS2 is expressed exclusively in erythroid cells. ALA is then exported to the cytosol, converted to coproporphyrinogen III (coPIII) by a series of enzymes, imported back into the mitochondrion, and converted to protoporphyrin IX (PPIX). Subsequently, ferrochelatase, an enzyme with an ISC as a prosthetic group, catalyzes the incorporation of Fe^2+^ into PPIX to form heme. Heme is then exported to the cytosol by the heme exporter feline leukemia virus type C receptor 1b (FLVRC1b) (Figure 1C). Mutations in ALAS2 or in SLC25A38 cause X-linked or congenital sideroblastic anemia, respectively, characterized by mitochondrial iron accumulation [24].

### 4.2. Biosynthesis of Iron-Sulfur Clusters (ISCs)

ISCs are prosthetic groups composed of iron and sulfide ions coordinated to form rhomboid (2Fe-2S) or cubane (4Fe-4S) structures. ISCs are of critical importance for cellular functions. They are involved in electron transport (as prosthetic groups of complex I, II, and III subunits), enzymatic activities (aconitase), and nucleic acid processing and repair. These cofactors also have a regulatory function, being involved in the modulation of gene expression, oxygen levels, and iron metabolism via iron regulatory proteins (IRPs) 1 and 2 (for details, see [25]). The biosynthesis of ISCs is a complex process involving several proteins and enzymes [26,27] (Figure 1C). Among them, frataxin plays a crucial role as an iron sensor in ISC and heme biosynthesis by binding iron and interacting with ferrochelatase. After ISC biosynthesis, the cofactor is transferred to target proteins by chaperones such as HSPA9 and glutaredoxin 5 (GLRX5). ISCs can also be exported to the cytosol via the ATP-binding cassette subfamily B member 7 (ABCB7) (Figure 1C). Mutations in genes encoding *GLRX5*, *HSPA9*, and the ISC exporter *ABCB7* cause rare forms of recessive or X-linked sideroblastic anemia [24].

## 5. Cellular Iron Homeostasis

Through the production of ISCs, mitochondria play a central role in coordinating cellular iron homeostasis. The expression and translation of proteins involved in iron import, utilization, storage, and export are tightly regulated by the Iron-Responsive Element (IRE)/iron regulatory protein (IRP) system (Figure 2). IRP1 and 2 are RNA-binding proteins that bind the IRE, an RNA stem-loop motif, in the untranslated region (UTR) of target genes. IRP binding blocks mRNA translation when the IRE is located in the 5’UTR, as in the case of ferritin H and L, FPN1, ALAS2, and HIF2α [28], or increases mRNA stability when the IRE is in the 3’UTR, as in the case of TFR1, DMT1, and the cell cycle regulator CDC14A [29] (Figure 2A). IRP1 can bind the ISC in its catalytic pocket, changing the protein’s function to that of a cytosolic aconitase. When the cellular iron concentration decreases (Figure 2A), ISC synthesis is reduced, and apo-IRP1 functions as an RNA-binding protein: by binding to the 5’- and 3’UTRs of target genes, it increases iron uptake and utilization while decreasing its storage and export [30]. The opposite occurs in conditions of increased intracellular iron concentration (Figure 2B). IRP2 has nearly 60% homology and overlaps in function with IRP1. However, its iron-mediated regulation differs from IRP1 since it does not bind ISCs. Instead, its stability is regulated by iron- and oxygen-dependent proteasomal degradation mediated by the binding of F-Box and Leucine-Rich Repeat Protein 5 (FBXL5) in complex with the SKP1-CUL1 ubiquitin ligase (Figure 2B). Interestingly, FBXL5 can bind iron in its C-terminal domain and the redox-sensitive 2Fe-2S cluster in its N-terminal domain: iron deficiency and/or changes in oxygen tension mediate FBXL5 destabilization and thus IRP2 stability [31,32] (Figure 2B).

## 6. Systemic Iron Homeostasis: The Hepcidin-Ferroportin Axis

### 6.1. Iron-Mediated Regulation

Iron is not only essential for various biological processes but also can be potentially toxic when in excess. Therefore, its levels should be tightly controlled and kept within a narrow window to meet the body’s iron needs and prevent the harmful effects of its accumulation. Since iron cannot be actively excreted, its concentration is regulated at the level of dietary absorption and release from stores, mainly in the liver and spleen.

The liver plays a central role in coordinating iron homeostasis in response to the body’s needs (Figure 3A). This function is performed by the hormone hepcidin, a 25-amino-acid peptide with four intramolecular disulfide bonds [33] that is secreted by hepatocytes [34] and regulates intestinal absorption and release from stores [35]. By binding and blocking FPN1, either by degradation [36] or by occlusion [37], hepcidin controls the iron flow into the blood (Figure 3A). Thus, the hepcidin–ferroportin axis prevents the formation of non-transferrin-bound iron (NTBI), which occurs when iron entering the bloodstream exceeds the binding capability of TF. Hepcidin levels are regulated mainly at the transcriptional level. Its expression is increased by elevated body iron, either as diferric-TF and/or as the liver iron concentration (Figure 3A), and is decreased when circulating and tissue iron levels drop below the safety threshold. This homeostatic modulation of hepcidin regulates the amount of FPN1 on the cell surface and thus iron release from duodenal enterocytes, splenic macrophages, and hepatocytes. Although hepcidin is also expressed by other cell types, such as inflammatory macrophages [38], epidermal cells [39], dendritic cells [40], and some cancer cells [41,42,43], only hepatocyte-derived hepcidin exerts systemic control of body iron levels [44]. In other tissues, such as the heart, hepcidin has an important cell-autonomous function in regulating cardiomyocyte iron concentration [45]. This finding suggests that hepcidin produced outside the liver, as in the kidney, the brain, and the placenta, has a local role in controlling cellular iron levels.

Molecularly, hepcidin expression in hepatocytes is regulated by the bone morphogenetic protein (BMP)-Son-of-Mother Against Decapentaplegic (SMAD) pathway [47] (Figure 3B), a signaling pathway involved in crucial biological processes, such as bone formation, development, inflammation, neuronal function, etc. The pathway is activated when a dimeric ligand interacts with plasma membrane BMP type I and II receptors, serine/threonine kinases, forming a hexameric complex [48]. In detail, BMP type II receptors are constitutively active and, in the presence of ligands, phosphorylate the intracellular domain of type I receptors. This domain is closed to the catalytic region of BMP type I receptors and responsible for the activation of their kinase activity. The activated BMP type I receptors then phosphorylate cytosolic SMAD1/5/8 that translocate into the nucleus in complex with the cargo protein SMAD4, where they function as transcription factors to induce the expression of BMP target genes [46]. In hepatocytes, the BMP receptors involved in hepcidin regulation are ALK2 and ALK3 (type I) [49] and ACVR2A and BMPR2 (type II) [50] (Figure 3B). Interestingly, although these BMP type II receptors have a redundant function in hepcidin regulation, since only the combined genetic deletions of both decrease hepcidin expression [50], ALK2 and ALK3 modulate hepatocyte BMP-SMAD signaling with a different magnitude. Indeed, *Alk2* deletion in hepatocytes only mildly affects hepcidin expression [49], whereas *Alk3* gene inactivation severely hampers hepcidin levels [51]. Accordingly, the mouse model characterized by the combined deletion of these BMP type I receptors is characterized by a more severe iron overload due to hepcidin deficiency than single mutants. Additional proteins, such as hemojuvelin (HJV), the second transferrin receptor TFR2, and the homeostatic iron regulator HFE positively modulate the BMP-SMAD pathway. HJV is a GPI-anchored protein highly expressed in hepatocytes, skeletal muscle, and the heart that functions as a BMP coreceptor. In hepatocytes, HJV is a positive regulator of hepcidin expression, and its mutations cause a juvenile form of hereditary hemochromatosis, a disease characterized by iron overload due to uncontrolled iron entry into the circulation consequent to reduced hepcidin production and FPN1 stabilization [52]. In the skeletal muscle, HJV prevents muscle atrophy, fibrosis, and reduced muscle force by counteracting dystrophic and age-related TGF-β1/Smad3 signaling activation [53]. TFR2, highly homologous to TFR1, is expressed mainly in hepatocytes and erythroid cells (Table 1). Although it binds diferric-TF as its counterpart, its binding affinity is considerably lower than (25 less) and its functional effect is opposite to that of TFR1. Indeed, while TFR1 binding to diferric-TF triggers receptor internalization and provides iron to the cell, TFR2-diferric-TF binding stabilizes the receptor on the cell surface. The latter process favors the transcriptional upregulation of hepcidin in hepatocytes and the regulation of erythropoietin (EPO) signaling through the EPO receptor in erythroid cells. Interestingly, TFR2 is considered the “iron sensor” that regulates hepcidin expression [54] and erythropoiesis according to serum iron availability. HFE is a positive regulator of hepcidin expression, likely because of its interaction with components of the BMP signaling pathway, such as TFR2 [55] and ALK3 [51]. Interestingly, in conditions of low serum iron, TFR1 interacts with HFE at the same site of diferric-TF binding. This binding is disrupted when serum iron increases to make or render HFE free to bind other partners and upregulate hepcidin [56]. Mutations in HFE cause the common adult-onset HH [52].

The ligands that upregulate hepcidin expression are BMP2 [57] and BMP6 [58,59] (Figure 3B). While the former is mainly involved in the regulation of basal hepcidin expression, the latter is upregulated by liver iron and is responsible for increased hepcidin expression in conditions of high body iron levels. BMP2 and BMP6 are expressed by liver sinusoidal endothelial cells (LSECs) [57,60]: however, how the LSEC “senses” iron levels and regulates BMP6 expression is still debated. It has been proposed that iron-dependent *Bmp6* upregulation in LSECs is cell-autonomous due to the activation of nuclear factor erythroid 2-related factor 2 (NRF2) [61], mediated by the intracellular internalization of NTBI [62]. However, LSECs in hemochromatosis *Hjv* KO mice express high *Bmp6* levels despite being iron-deficient due to FPN1 stabilization by low hepatocyte hepcidin production, suggesting that other mechanisms are involved in *Bmp6* upregulation [63]. Interestingly, a recent paper demonstrates that in the presence of iron, hepatocytes secrete a still-unknown protein that functions as a BMP6 activator, suggesting that cell-to-cell communication between hepatocytes and LSECs is essential for iron sensing and *Bmp6* regulation [64]. Thus, hepcidin regulation by BMPs represents a beautiful example of the importance of intraorgan crosstalk between different cell types, such as hepatocytes and LSECs.

Although the critical role of the BMP-SMAD pathway in hepatocytes in the regulation of hepcidin expression has been proven, several molecular details are yet to be unraveled. Hepcidin expression is controlled by ALK2 and ALK3, the BMP type I receptors that regulate the signaling pathway with a different power/intensity. Why are two receptors needed, and how is their activation regulated? Additionally, hepcidin expression is impaired when HJV, HFE, and TFR2 are mutated, as in hereditary hemochromatosis. HJV is a BMP coreceptor. However, how the hemochromatosis proteins HFE and TFR2 regulate hepcidin expression is unclear. Recent data have shown that HFE signals mainly via the BMP type I receptor ALK3 and modulates hepcidin by interacting with TFR1, but the mechanism is obscure. Furthermore, TFR2, highly homologous to TFR1, can bind diferric-TF, and this binding stabilizes the receptor on the hepatocyte surface. However, how this stabilization signals through the BMP-SMAD pathway for hepcidin upregulation is presently unknown. Moreover, how does the liver sense plasma and tissue iron levels and regulate hepcidin expression accordingly? How does iron regulate BMP6 expression? According to recent data, the crosstalk between hepatocytes and LSECs is crucial, but the molecular mechanism/s of this crosstalk remain unclear.

In iron deficiency, hepcidin should be downregulated to allow iron entry into the circulation through FPN1. Both decreased diferric-TF and reduced BMP6 levels contribute to BMP-SMAD pathway/hepcidin downregulation [46]. In addition, a recently identified inhibitor of the BMP pathway, the immunophilin FKBP12, binds to ALK2 and blocks the activation of this receptor in the absence of the ligand [65]. However, the main hepcidin inhibitor is type II transmembrane serine protease 6 (TMPRSS6 or matriptase-2), whose mutations are associated with a rare genetic disease named Iron-Refractory Iron Deficiency Anemia (IRIDA), characterized by iron deficiency and anemia due to constitutively and inappropriately high hepcidin levels [66].

It has been shown that TMPRSS6 cleaves HJV in vitro [67], and that the genetic inactivation of *Hjv* in *Tmprss*6 KO mice reverts the IRIDA phenotype in vivo, suggesting that HJV is the endogenous substrate. However, recent in vitro studies suggest that other components of the BMP pathway are cleaved by TMPRSS6 [68], and, more intriguingly, it seems that the protease activity of TMPRSS6 is not needed for its inhibitory capacity [69]. Overall, these findings add a new layer of complexity to the mechanisms of hepcidin regulation.

### 6.2. Inflammation-Mediated Regulation

Hepcidin was initially identified by in vitro studies as a novel hepatic antimicrobial peptide [33,34] with a defensin-like structure because of its ability to counteract fungal and bacterial growth [34]. However, in vivo, the principal antimicrobial activity of hepcidin is indirect and mediated by its effect on decreasing FPN1 activity that, causing cellular iron retention and hypoferremia, reduces iron availability for microbial growth (Figure 3C). Because hepcidin is an acute-phase protein [70], in conditions of infection/inflammation, its expression is increased by proinflammatory cytokines, mainly IL6 and IL1β, through the JAK2-STAT3 pathway [71] (Figure 3D). Hepcidin upregulation represents the first line of defense to counteract the growth of invading extracellular pathogens. When infection/inflammation persists, hypoferremia due to chronic hepcidin upregulation causes anemia due to the reduced iron supply for hemoglobin synthesis and RBC production, a disorder known as anemia of chronic diseases (ACD) or anemia of inflammation (AI) [72,73]. However, in rare cases, hepcidin also works as a direct antimicrobial peptide, as in severe skin infections such as necrotizing fasciitis due to group A Streptococcus, where it promotes the production of the CXCL1 chemokine, essential for neutrophil recruitment to recover from the infection, by keratinocytes [39].

### 6.3. Erythroid-Mediated Regulation

Terminal erythropoiesis, which generates red blood cells from erythroid precursors under erythropoietin (EPO) stimulation, highly depends upon the serum iron concentration [74]. In mammals, about 80% of circulating iron is mainly consumed by erythroid cells in the form of diferric-TF for hemoglobin synthesis. Since the 1–2 mg of iron absorbed per day is insufficient to cope with erythron needs, which are around 25 mg/day, the body has developed several mechanisms to guarantee the availability of this metal for erythropoiesis.

First, iron is recycled through the phagocytosis of senescent red blood cells by reticuloendothelial macrophages in the spleen (and the liver in some conditions) that degrade hemoglobin and deliver iron to plasma transferrin via FPN1 [75].

Second, when erythropoiesis is increased, such as in hypoxic conditions, following bleeding, or in severe anemia, liver hepcidin expression is downregulated to stabilize FPN1 on the cell surface [76], favoring iron entry into the circulation from duodenal enterocytes, macrophages, and hepatocytes. This is a beautiful example of interorgan crosstalk and underlines how maintaining serum iron levels is essential for the efficient production of red blood cells.

How does the erythron signal its iron needs to the body?

The existence of the so-called “erythroid regulator”, a soluble molecule released by developing erythroblasts and able to increase intestinal iron absorption, was postulated several years ago by Clem Finch [77]. This protein should be produced by proliferating erythroblasts, since blocking cell proliferation through cytostatic agents blunted hepcidin downregulation following erythropoiesis stimulation [76]. Several molecules have been proposed as “erythroid regulators”, such as growth differentiation factor 15 (GDF15), a member of the Tumor Growth Factor β family produced and released by erythroblasts in response to EPO stimulation [78], and Twisted Gastrulation Protein Homolog 1 (TWSG1), an antagonist of the BMP pathway, highly expressed in early erythroblasts and thalassemic mice [79]. However, their role as potential hepcidin inhibitors in vivo remains unclear. The “long sought” erythroid regulator Erythroferrone (ERFE) was identified by searching for erythroid-specific mRNAs encoding secreted proteins whose expression was increased by erythropoiesis expansion induced by EPO injection or bleeding [80] (Figure 4). Thus, when EPO is increased by anemia/hypoxia, *ERFE* expression by erythroid progenitors is upregulated, and the protein is released into the circulation. Then, ERFE inhibits hepcidin expression by binding and sequestering several BMPs, such as BMP5, BMP7, and the iron-regulated BMP6 [81]. FPN1 stabilization on the cell surface increases iron entry into the bloodstream to sustain hemoglobin synthesis in developing erythroblasts. In the basal state, *Erfe* KO mice have hepcidin levels comparable to their wild-type littermates. However, in conditions of increased EPO, *Erfe* KO mice cannot properly downregulate hepcidin as wild-type mice can and are delayed in Hb recovery due to their inability to increase iron entry into the circulation [80].

*Erfe* is upregulated in erythroid tissues in all conditions characterized by increased serum EPO and expanded erythropoiesis, such as iron deficiency anemia and hypoxia [80], but also in diseases characterized by ineffective erythropoiesis, such as beta-thalassemia [82]. Here, excessive ERFE production causes secondary iron overload, as also observed in mice overexpressing *Erfe* [83]. However, preclinical models demonstrate that ERFE is relevant in hepcidin downregulation in conditions of acute stress erythropoiesis, but not in chronic conditions [84], where other mechanism/s are likely involved. For example, platelet-derived growth factor-BB (PDGF-BB), a secreted protein released by endothelial cells, macrophages, and platelets in response to hypoxia, may contribute to hepcidin downregulation [85] (Figure 4).

## 7. Concluding Remarks

In recent years, significant advances in understanding how iron homeostasis is regulated and the clinical implications of its deregulation have been attained. Nevertheless, several questions still need to be answered. Understanding the detailed molecular mechanisms of BMP-SMAD pathway regulation by plasma and tissue iron and how the hepatocyte responds to a combination of different signals will increase our understanding of how systemic iron homeostasis is maintained in physiological conditions and how it is deranged in disorders due to hepcidin deregulation, exemplified by hemochromatosis, thalassemia, IRIDA, and inflammatory disorders. The knowledge of the basic mechanisms involved in the homeostatic regulation of iron metabolism will help in the identification of new therapeutic targets for these diseases.

## Figures and Tables

**Figure 1 ijms-24-03995-f001:**
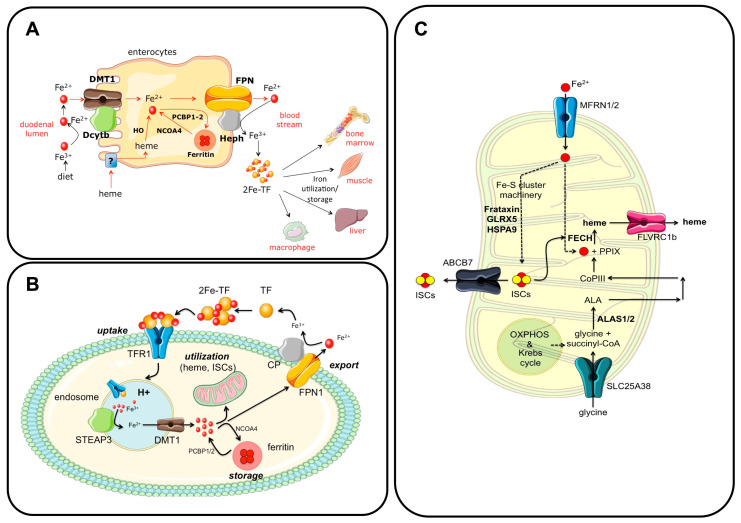
Regulation of iron absorption and utilization. (**A**) Iron homeostasis in enterocytes. The main players are depicted in the figure. Briefly, dietary iron is absorbed through DMT1 and DCytB, expressed on the apical side of enterocytes. In the cytosol, excess iron can be stored in ferritin or exported to the bloodstream by ferroportin (FPN1), whose function is coupled with the ferroxidase activity of hephaestin (HEPH). Iron circulates bound to the glycoprotein transferrin (Diferric-TF or 2Fe-TF), and it is primarily utilized by erythroid cells. Excess iron is stored in the liver and splenic macrophages. (**B**) The iron cycle. Diferric-transferrin (2Fe-TF) is taken up through transferrin receptor 1 (TFR1) binding and clathrin-mediated endocytosis. In the acidic endosome, Fe^3+^ dissociates from TF, is reduced to Fe^2+^ by six-transmembrane epithelial antigen of prostate member 3 (STEAP3), and exported to the cytosol by divalent metal transporter 1 (DMT1). TF-TFR1 are recycled back to the cell surface, where the neutral pH of the extracellular environment favors their dissociation. In the cytosol, iron can be stored in ferritin nanocages thanks to the chaperones PCBP1/2 and released when needed through NCOA4 in a process named ferritinophagy or imported into mitochondria. Excess iron is exported outside the cell through FPN1, whose function is coupled with the ferroxidase activity of ceruloplasmin (CP). Fe^3+^ is efficiently bound to TF for its distribution. (**C**) Mitochondrial iron utilization. Iron enters the mitochondria through mitoferrin 1 and 2 (MFRN1/2), and participates in iron–sulfur cluster (ISC) and heme biosynthesis. In the mitochondria, ISCs control the enzymes of the respiratory chain and Krebs cycle and ferrochelatase (FECH), the rate-limiting enzyme that synthesizes heme from protoporphyrin IX (PPIX) and iron. ISCs are exported to the cytosol by ABCB7, whereas heme is exported to the cytosol by FLVCR1b. Panels B and C: adapted with permission from Ref. [6]. License date: 15 February 2023; Elsevier.

**Figure 2 ijms-24-03995-f002:**
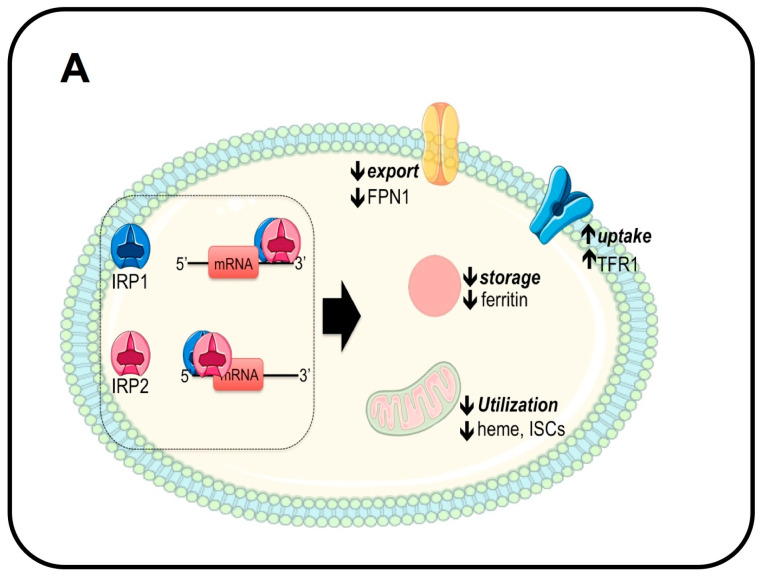
Mechanism of cellular iron homeostasis by the IRP/IRE system. (**A**) In conditions of decreased intracellular iron concentration, iron regulatory proteins 1 and 2 (IRP1 and IRP2) function as RNA-binding proteins and reduce the translation of ferritin and ferroportin (FPN1) and decrease mitochondrial iron utilization by binding the Iron-Responsive Element (IRE) in the 5′UTR. Conversely, IRP1 and 2 increase iron uptake by binding the IRE in the 3′UTR of TFR1 and DMT1. (**B**) When intracellular iron levels increase, IRP1 binding to iron–sulfur clusters (ISCs) converts the RNA-binding protein into aconitase, whereas IRP2 is degraded by the proteasome via a mechanism dependent on iron, ISC, and oxygen levels. Thus, ferritin and FPN1 are translated to favor iron storage and export, and the utilization of iron by the mitochondria is increased. Conversely, TFR1 and DMT1 mRNAs are reduced. Panels A and B: adapted with permission from Ref. [6]. License date: 15 February 2023; Elsevier.

**Figure 3 ijms-24-03995-f003:**
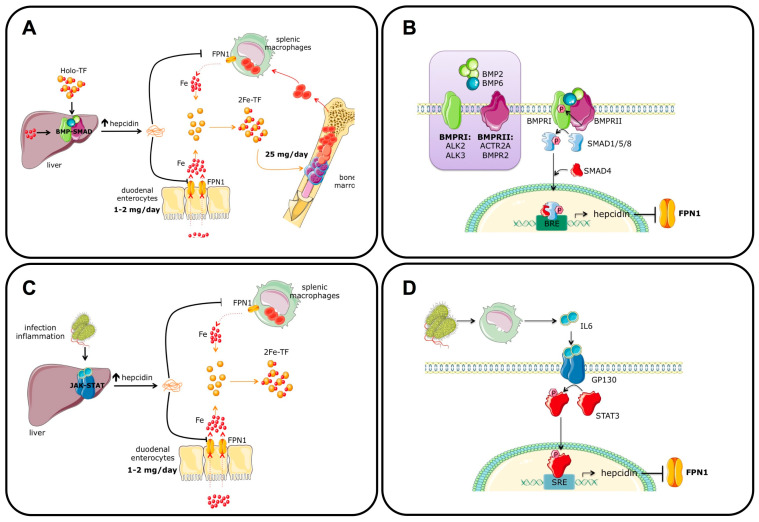
Regulation of systemic iron homeostasis by iron and inflammation. Dietary iron is absorbed by duodenal enterocytes and enters the circulation through the iron exporter ferroportin (FPN1). Iron circulates bound to transferrin and is internalized through the TF-TFR1 mechanism (Figure 1) by organs and tissues, mainly by erythroid cells for hemoglobin synthesis. Excess iron cannot be actively excreted and is safely stored in the liver and in reticuloendothelial macrophages of the spleen. Phagocytosis of senescent red blood cells and hemoglobin degradation by macrophages provide most of the iron utilized for hemoglobin synthesis. The liver regulates the amount of circulating iron to prevent the formation of non-transferrin-bound iron (NTBI) through hepcidin, a hormone that binds and blocks FPN1 function. Hepcidin expression and synthesis are increased by iron (**A**) and inflammation (**C**) to reduce iron entry into the bloodstream. (**B**) Mechanism of hepcidin regulation by the BMP-SMAD pathway. In the presence of BMP2/6, the constitutively active BMP type II receptors (BMPRII) phosphorylate and activate BMP type I receptors (BMPRI). BMPRIs then phosphorylate regulatory SMAD proteins such as SMAD1/5/8. Once phosphorylated, they interact with SMAD4 for nuclear translocation, where the complex recognizes BMP-Responsive Elements (BREs) in the promoter region of hepcidin. (**D**) Macrophage-derived IL6 binds to its receptor GP130 in hepatocytes and activates the JAK2-STAT3 signaling pathway: phosphorylated STAT3 translocates into the nucleus to bind the STAT3-responsive element (SRE) on the hepcidin promoter and activates its expression. Panels (**B**,**D**): adapted with permission from Ref. [46]. License date: 15 February 2023; Elsevier.

**Figure 4 ijms-24-03995-f004:**
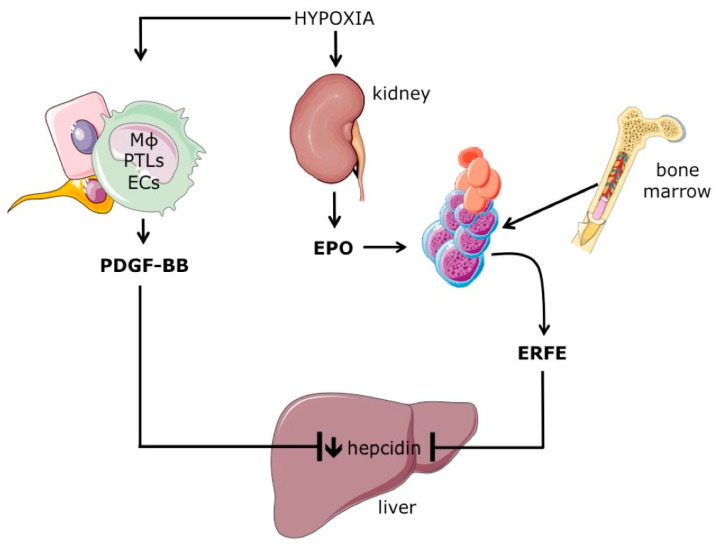
Mechanisms of hepcidin inhibition by increased erythropoiesis. Systemic hypoxia increases erythropoietin (EPO) production by the kidney. EPO, by stimulating erythroid cell proliferation and differentiation, also increases the expression the Erythroferrone (ERFE), a secreted protein that inhibits the expression of hepcidin by hepatocytes by sequestering BMP ligands. Hypoxia is also sensed by other cells, such as macrophages (MΦ), platelets (PTLs), and endothelial cells (ECs), that increase the production of serum platelet-derived growth factor-BB (PDGF-BB), which contributes to hepcidin downregulation. Adapted with permission from Ref. [6]. License date: 15 February 2023; Elsevier.

**Table 1 ijms-24-03995-t001:** Comparison between TFR1 and TFR2.

Features	TFR1	TFR2
Expression	Ubiquitous	Restricted to hepatocytes, erythroid cells, and osteoblasts
Diferric-TF binding	Yes	Yes (25 lower affinity than TFR1)
Membrane stabilization by Diferric-TF	Yes	Yes
3′IRE	Yes	No
dimerization	Yes	Yes
Interacting proteins	TF and HFE	TF, HFE, HJV, EPOR
Inactivation	Severe IDA in mouse	IO (type III HH) in human and mouse

TFR1: transferrin receptor 1; TF: transferrin; IRE: Iron-Responsive Element; HFE: homeostatic iron regulator; HJV: hemojuvelin; EPOR: erythropoietin receptor; IDA: iron deficiency anemia; IO: iron overload; HH: hereditary hemochromatosis.

## Data Availability

Not applicable.

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
