# Peer review of "Managing the Dual Nature of Iron to Preserve Health"

_ijms, 2023, doi:10.3390/ijms24043995_

Round 1

Reviewer 1 Report

Iron as essential element in living organisms (regulation of iron absorbtion, utilizarion and significance in normal and pathological conditions) is well presented at the cellular and molecular levels.

Further reviews could clarify the problems at different stages of bone marrow and spleen erythropoiesis. From  the technical point of view I could recommend little corrections in phrases, which are very long and complicated. I think that the number of Keywords is unsufficient and could be enlarged.

Best regards!

Author Response

We thank the reviewer for the useful comments. We have modified the text accordingly.

Reviewer 2 Report

I have read an interesting review by Laura Silvestri et al. about the amazing properties of iron. It is an interesting topic, but the authors should revise the manuscript in a more comprehensive manner, bearing in mind that this is a literature review.

Here are my comments regarding this paper:

Ü  Major comments:

-       Iron is an extremely studied molecule, and authors should add more information to this manuscript to be a comprehensive review (the manuscript has only 70 references)

-       Introduction section is extremely scarce and should be revised.

-  Subsection 7 is about the role of hepcidin in regulation of iron homeostasis, and there are plenty of research on this topic. Please try to add more information here.

-       Conclusion section is too extensive. It needs to be more concise and some of the information needs to be moved to the main body or the manuscript.

Ü  Minor comments:

-       Abbreviations should be explained at first appearance in the text, irrespective of the abstract section (e.g., DNA, HFE, and so on)

-       Please try to rewrite the first sentence of the introduction section not to start with “because”.

-       All tables should have a footer, and all the abbreviation within the table should be explained here.

-       Not sure about line 345-348. Maybe you can rewrite this part.

-       Are all the figures designed by the authors, or is there a source of inspiration? If there is, it should be mentioned next to the figures.

Author Response

We thank the reviewer for these suggestions. The text has been improved as requested and amended where needed.